# Taxonomy and Phylogeny of the Wood-Inhabiting Fungal Genus *Hyphoderma* with Descriptions of Three New Species from East Asia

**DOI:** 10.3390/jof7040308

**Published:** 2021-04-16

**Authors:** Qian-Xin Guan, Chang-Lin Zhao

**Affiliations:** 1Key Laboratory for Forest Resources Conservation and Utilization in the Southwest Mountains of China, Ministry of Education, Southwest Forestry University, Kunming 650224, China; zongtongkai@im.ac.cn; 2College of Biodiversity Conservation, Southwest Forestry University, Kunming 650224, China

**Keywords:** corticioid fungi, *Hyphoderma*, hyphodermataceae, molecular systematics, Yunnan Province

## Abstract

Three new wood-inhabiting fungi, *Hyphoderma crystallinum*, *H. membranaceum,* and *H. microporoides* spp. nov., are proposed based on a combination of morphological features and molecular evidence. *Hyphoderma crystallinum* is characterized by the resupinate basidiomata with smooth hymenial surface scattering scattered nubby crystals, a monomitic hyphal system with clamped generative hyphae, and numerous encrusted cystidia present. *Hyphoderma membranaceum* is characterized by the resupinate basidiomata with tuberculate hymenial surface, presence of the moniliform cystidia, and ellipsoid to cylindrical basidiospores. *Hyphoderma microporoides* is characterized by the resupinate, cottony basidiomata distributing the scattered pinholes visible using hand lens on the hymenial surface, presence of halocystidia, and cylindrical to allantoid basidiospores. Sequences of ITS+nLSU rRNA gene regions of the studied samples were generated, and phylogenetic analyses were performed with maximum likelihood, maximum parsimony, and Bayesian inference methods. These phylogenetic analyses showed that three new species clustered into *Hyphoderma*, in which *H. crystallinum* was sister to *H. variolosum*, *H. membranaceum* was retrieved as a sister species of *H. sinense*, and *H. microporoides* was closely grouped with *H. nemorale*. In addition to new species, map to show global distribution of *Hyphoderma* species treated in the phylogenetic tree and an identification key to Chinese *Hyphoderma* are provided.

## 1. Introduction

Fungi are an ecologically important branch of the tree of life based on its distinct and diverse characteristics, in which these organisms play a vital role in ecosystems as diverse as soil, forest, rocks, and ocean, but their roles are primarily enacted behind the scenes, literally as hidden layers within their substrate [1]. On the basis of the nature of their intertwined partners in numerous symbiotic interactions, they have mostly marched via stepwise codiversification with the plants [2]. Fungi have evolved numerous strategies to degrade hard-to-digest substrates for outcompeting with other microbes, while combating competitors using an arsenal of bioactive metabolites, such as the familiar antibiotics, ethanol, and organic acids [3]. Taxonomy plays a central role in understanding the diversity of life, discovering into systems of names that capture the relationships between species, and translating the products of biological exploration [4]. Despite the early embrace of the molecular systematics by mycologists, both the discovery and classification of fungi are still in great flux, particularly among the more basal branches of the tree, in which the true diversity is only now coming to light from genomic analyses and environmental DNA surveys [1]. The researches revealed that perhaps less than 5% of the estimated two to four million species have been formally described, therefore, the hidden and microscopic nature of many fungi also means that their diversity is undersampled [5,6].

The genus *Hyphoderma* Wallr. is one of the most important fungal groups because of its key role in the carbon cycle and being the most efficient wood decomposers in the forest ecosystem [7]. This genus is typified by *H. setigerum* (Fr.) Donk [8]. *Hyphoderma* is characterized by the resupinate to effuse-reflexed basidiomata with ceraceous consistency, and smooth to tuberculate or hydnoid hymenophore and a monomitic hyphal structure (rarely dimitic) with clamp connections on generative hyphae, presence of cystidia or not, basidia suburniform to subcylindrical and cylindrical, ellipsoid to subglobose, smooth, thin-walled basidiospores [9,10]. Currently, about 100 species have been accepted in *Hyphoderma* worldwide [8,11,12,13,14,15]. Index Fungorum (http://www.indexfungorum.org; accessed on 16 April 2021) and MycoBank (https://www.mycobank.org; accessed on 16 April 2021) register 192 specific and infraspecific names in *Hyphoderma*.

Molecular systematics covering *Hyphoderma* revealed the classification of corticioid fungi and showed that *H. obtusum* J. Erikss. and *H. setigerum* clustered into Meruliaceae Rea and then grouped with *Hypochnicium polonense* (Bres.) Å. Strid, based on the internal transcribed spacer (ITS) regions and the large subunit nuclear ribosomal RNA gene (nLSU) sequences [16]. Telleria et al. [17] discussed the relationships between *Hyphoderma* and *Peniophorella* P. Karst., in which some species from *Hyphoderma* and *Peniophorella* are grouped and they proposed a new species, *H. macaronesicum* Tellería et al. The research on studying *Hyphoderma setigerum* complex showed that *H. pinicola* Yurch. and Sheng H. Wu represented a fifth species in this complex, which revealed that this complex was a white-rot wood-decaying corticoid fungal species and occurred worldwide from tropical to temperate regions [18]. A revised family-level classification of the Polyporales revealed that four *Hyphoderma* species nested into the residual polyporoid clade belonging to Hyphodermataceae, and then, they were grouped with three related genera *Meripilus* P. Karst., *Physisporinus* P. Karst., and *Rigidoporus* Murrill [19]. Chinese *Hyphoderma* species were compared with closely related taxa, and two new species were proposed, *H. fissuratum* C.L. Zhao and X. Ma and *H. mopanshanense* C.L. Zhao [15].

In this study, three undescribed species of corticioid fungi were collected from Yunnan Province, China. Morphological characteristics and molecular phylogenetic analyses of combined ITS+nLSU rRNA sequences supported the recognition of three new species within *Hyphoderma*.

## 2. Materials and Methods

### 2.1. Morphology

The studied specimens are deposited at the herbarium of Southwest Forestry University (SWFC), Kunming, Yunnan Province, China. Macromorphological descriptions are based on field notes and photos captured in the field and laboratory. Color terminology follows Petersen [20]. Micromorphological data were obtained from the dried specimens, which were observed under a light microscope following Dai [21]. The following abbreviations are used: KOH = 5% potassium hydroxide water solution, CB = Cotton Blue, CB– = acyanophilous, IKI = Melzer’s reagent, IKI– = both inamyloid and indextrinoid, L = mean spore length (arithmetic average for all spores), W = mean spore width (arithmetic average for all spores), Q = variation in the L/W ratios between the specimens studied, n = a/b (number of spores (a) measured from given number (b) of specimens).

### 2.2. Molecular Phylogeny

CTAB rapid plant genome extraction kit-DN14 (Aidlab Biotechnologies Co., Ltd., Beijing, China) was used to obtain genomic DNA from dried specimens, according to the manufacturer’s instructions [22]. ITS region was amplified with primer pair ITS5 and ITS4 [23]. Nuclear nLSU region was amplified with primer pair LR0R and LR7 (http://lutzonilab.org/nuclear-ribosomal-dna/; accessed on 16 April 2021). The PCR procedure for ITS was as follows: initial denaturation at 95 °C for 3 min; followed by 35 cycles at 94 °C for 40 s, 58 °C for 45 s, and 72 °C for 1 min; and a final extension of 72 °C for 10 min. The PCR procedure for nLSU was as follows: initial denaturation at 94 °C for 1 min; followed by 35 cycles at 94 °C for 30 s, 48 °C for 1 min, and 72 °C for 1.5 min; and a final extension of 72 °C for 10 min. The PCR products were purified and directly sequenced at Kunming Tsingke Biological Technology Limited Company, Kunming, Yunnan Province, China. All newly generated sequences were deposited in NCBI GenBank (Table 1).

Sequences were aligned in MAFFT 7 (https://mafft.cbrc.jp/alignment/server/; accessed on 10 April 2021) using the “G-INS-i” strategy for ITS+nLSU and manually adjusted in BioEdit [32]. The dataset was aligned first and then, ITS and nLSU sequences were combined with Mesquite. Alignment datasets were deposited in TreeBASE (submission ID 27983). *Climacocystis borealis* (Fr.) Kotl. and Pouzar and *Diplomitoporus crustulinus* (Bres.) Domański were selected as an outgroup for phylogenetic analysis of ITS+nLSU phylogenetic tree (Figure 1) following a previous study [19].

Maximum parsimony analysis was applied to the combined (ITS+nLSU) dataset. Its approaches followed previous study [22], and the tree construction procedure was performed in PAUP* version 4.0b10 [33]. All characters were equally weighted and gaps were treated as missing data. Trees were inferred using the heuristic search option with TBR branch swapping and 1000 random sequence additions. Max-trees were set to 5000, branches of zero length were collapsed, and all parsimonious trees were saved. Clade robustness was assessed using bootstrap (BT) analysis with 1000 replicates [34]. Descriptive tree statistics: tree length (TL), consistency index (CI), retention index (RI), rescaled consistency index (RC), and homoplasy index (HI) were calculated for each maximum parsimonious tree generated. Datamatrix was also analyzed using maximum likelihood (ML) approach with RAxML-HPC2 through the CIPRES Science Gateway (www.phylo.org; accessed on 8 April 2021) [35]. Branch support (BS) for ML analysis was determined by 1000 bootstrap replicates.

MrModeltest 2.3 [36] was used to determine the best-fit evolution model for each data set for Bayesian inference (BI). BI was calculated with MrBayes 3.1.2 [37]. Four Markov chains were run for 2 runs from random starting trees for 6 million generations for ITS+nLSU (Figure 1). The first one-fourth of all generations was discarded as burn-in. The majority rule consensus tree of all remaining trees was calculated. Branches were considered as significantly supported if they received maximum likelihood bootstrap value (BS) > 70%, maximum parsimony bootstrap value (BT) > 70%, or Bayesian posterior probabilities (BPP) > 0.95.

## 3. Results

### 3.1. Molecular Phylogeny

The ITS+nLSU dataset (Figure 1) included sequences from 78 fungal specimens representing 44 taxa. The dataset had an aligned length of 2086 characters, of which 1245 characters are constant, 127 are variable and parsimony-uninformative, and 714 are parsimony-informative. Maximum parsimony analysis yielded 5000 equally parsimonious trees (TL = 3441, CI = 0.3787, HI = 0.6213, RI = 0.7178, RC = 0.2718). Best model for the ITS+nLSU dataset estimated and applied in the Bayesian analysis was GTR+I+G (lset nst = 6, rates = invgamma; prset statefreqpr = dirichlet (1,1,1,1)). Bayesian analysis and ML analysis resulted in a similar topology to MP analysis with an average standard deviation of split frequencies = 0.007698 (BI).

The phylogram inferred from ITS+nLSU sequences (Figure 1) demonstrated that three new species are clustered into genus *Hyphoderma*, in which *H. crystallinum* was sister to *H. variolosum* Boidin, Lanq. and Gilles, *H. membranaceum* was retrieved as a sister species of *H. sinense* C.L. Zhao and Q.X. Guan, and *H. microporoides* was closely grouped with *H. nemorale* K.H. Larss. (100% BS, 100% BP, and 1.00 BPP).

### 3.2. Taxonomy

***Hyphoderma crystallinum*** C.L. Zhao and Q.X. Guan, sp. nov. Figure 2 and Figure 3.

MycoBank no.: MB 839276.

**Holotype**—China, Yunnan Province, Puer, Jingdong County, the Forest of Pineapple, E 100°48′, N 24°21′, 2113 m asl., on fallen angiosperm branch, leg. C.L. Zhao, 4 January 2019, C.L. Zhao 9338 (SWFC).

**Etymology—*crystallinum*** (Lat.): referring to the numerous and larger crystals on the hymenial surface.

**Fruiting body**—Basidiomata annual, resupinate, adnate, without odor and taste when fresh, membranaceous when fresh, becoming hard membranaceous upon drying, and up to 15 cm long, 3 cm wide, and 30–100 µm thick. Hymenial surface white to pale gray when fresh, pale gray to slightly cream upon drying, with scattered nubby crystals. Margin sterile indistinct and white.

**Hyphal system**—Monomitic, generative hyphae with clamp connections, colorless, thin-walled, frequently branched, interwoven, 2–3.5 µm in diameter, IKI-, CB-; tissues unchanged in KOH.

**Hymenium**—Cystidia of two types: (1) tubular cystidia, colorless, thin-walled, 32–51 µm × 5–10 µm and (2) encrusted cystidia, numerous, colorless, encrusted by crystals, 14–46 µm × 4–11 µm. Basidia clavate to subcylindrical, slightly constricted in the middle to somewhat sinuous, with 4 sterigmata and a basal clamp connection, 21.5–31 µm × 6–8.5 µm.

**Spores**—Basidiospores allantoid, colorless, thin-walled, smooth, with oil drops inside, IKI-, CB-, (10.5–)11–14.5(–15) µm × 4–5.5(–6) µm, L = 12.99 µm, W = 4.81 µm, Q = 2.47–2.98 (n = 90/3).

**Additional specimens examined**—China, Yunnan Province, Puer, Jingdong County, the Forest of Pineapple, E 100°48′, N 24°21′, 2113 m asl., on fallen angiosperm branch, leg. C.L. Zhao, 4 January 2019, C.L. Zhao 9374 (SWFC); Dali, Nanjian County, Lingbaoshan National Forestry Park, E 100°30′, N 24°46′, 1963 m asl., on fallen angiosperm branch, leg. C.L. Zhao, 9 January 2019, C.L. Zhao 10224 (SWFC); Wenshan, Funing County, Guying village, E 105°35′, N 23°36′, 976 m asl., on fallen angiosperm branch, leg. C.L. Zhao, 20 January 2019, C.L. Zhao 11723 (SWFC); Wenshan, Xichou County, Jiguanshan Forestry Park, E 103°46′, N 23°33′, 1670 m asl., on fallen angiosperm trunk, leg. C.L. Zhao, 22 July 2019, C.L. Zhao 15841 (SWFC); Honghe, Pingbian County, Daweishan National Nature Reserve, E 103°35′, N 22°53′, 1990 m asl., on fallen angiosperm branch, leg. C.L. Zhao, 3 August 2019, C.L. Zhao 18459 (SWFC).

***Hyphoderma membranaceum*** C.L. Zhao and Q.X. Guan sp. nov. Figure 4 and Figure 5.

MycoBank no.: MB 839278.

**Holotype**—China, Yunnan Province, Chuxiong, Zixishan Forestry Park, E 101°24′, N 25°01′, 2356 m asl., on fallen angiosperm branch, leg. C.L. Zhao, 1 July 2018, C.L. Zhao 6971 (SWFC).

**Etymology—*membranaceum*** (Lat.): referring to the membranous hymenophore.

**Fruiting body**—Basidiomata annual, resupinate, adnate, membranous, without odor and taste when fresh, and up to 15 cm long, 2 cm wide, and 30–100 µm thick. Hymenial surface tuberculate, white to pale gray when fresh, pale gray to cream on drying, with cracking. Margin sterile, narrow, and gray.

**Hyphal system**—Monomitic, generative hyphae with clamp connections, colorless, thin-walled, frequently branched, interwoven, 2.5–4.5 µm in diameter; IKI-, CB-; tissues unchanged in KOH.

**Hymenium**—Cystidia moniliform, thin-walled, 28–60 µm × 6.5–10.5 µm; basidia clavate to subcylindrical, slightly constricted in the middle to somewhat sinuous, with 4 sterigmata and a basal clamp connection, 21.5–31 µm × 5–7.5 µm.

**Spores**—Basidiospores ellipsoid to cylindrical, colorless, thin-walled, smooth, with irregular vacuole inside, IKI-, CB-, (10.5–)11–13.5(–14) µm × 4.5–5.5(–6) µm, L = 12.52 µm, W = 5.18 µm, Q = 2.42 (*n* = 60/2).

**Additional specimens examined**—China, Yunnan Province, Puer, Zhenyuan County, Heping Town, Liangzizhai, E 101°25′, N 23°56′, 2246 m asl., on fallen angiosperm branch, leg. C.L. Zhao, 15 Jan 2018, C.L. Zhao 5844 (SWFC).

***Hyphoderma microporoides*** C.L. Zhao and Q.X. Guan sp. nov. Figure 6 and Figure 7.

MycoBank no.: MB 839277.

**Holotype**—China, Yunnan Province, Chuxiong, Zixishan Forestry Park, E 101°24′, N 25°01′, 2313 m asl., on fallen angiosperm trunk, leg. C.L. Zhao, 30 June 2018, C.L. Zhao 6857 (SWFC).

**Etymology—*microporoides*** (Lat.): referring to the scattered pinholes on the hymenophore that are visible under hand lens.

**Fruiting body**—Basidiomata annual, resupinate, adnate, without odor and taste when fresh, cottony when fresh, fragile upon drying, and up to 22 cm long, 2.5 cm wide, and 50–100 µm thick. Hymenial surface smooth with scattered pinholes visible under hand lens, cream to pale buff when fresh, and slightly buff upon drying. Margin sterile, indistinct, and white to cream.

**Hyphal system**—Monomitic, generative hyphae with clamp connections, colorless, thin-walled, frequently branched, interwoven, 3–5 µm in diameter, IKI-, CB-; tissues unchanged in KOH.

**Hymenium**—Halocystidia capitate, thin-walled, smooth, 18–51 µm × 4.5–7 µm; basidia clavate, constricted, somewhat sinuous, with 4 sterigmata and a basal clamp connection, 18.5–29.5 µm × 5–7 µm.

**Spores**—Basidiospores cylindrical to allantoid, colorless, thin-walled, smooth, with oil drops inside, IKI-, CB-, 8.5–10(–10.5) µm × 2.5–3.5(–4) µm, L = 9.29 µm, W = 3.24 µm, Q = 2.87 (*n* = 30/1).

**Additional specimens examined**—China, Yunnan Province, Puer, Jingdong County, Taizhong Town, Ailaoshan Ecological Station, E 100°56′, N 24°29′, 1938 m asl., on fallen angiosperm branch, leg. C.L. Zhao, 25 August 2018, C.L. Zhao 8695 (SWFC).

## 4. Discussion

In the present study, three new species, *Hyphoderma crystallinum*, *H. membranaceum,* and *H. microporoides* are described based on phylogenetic analyses and morphological characteristics.

Phylogenetically, the family-level classification of the Polyporales (Basidiomycota) amplified nLSU, nITS, and rpb1 genes across the Polyporales, was employed, in which four species *Hyphoderma macaronesicum*, *H. medioburiense* (Burt) Donk, *H. mutatum* (Peck) Donk, and *H. setigerum*, nested into family Hyphodermataceae within the residual polyporoid clade [19]. In the present study, three new taxa clustered into *Hyphoderma*, in which *Hyphoderma crystallinum* was sister to *H. variolosum*, *H. microporoides* grouped closely with *H. nemorale,* and *H. membranaceum* grouped with *H. sinense* and *H. transiens* (Bres.) Parmasto (Figure 1). However, morphologically, *H. variolosum* differs from *H. crystallinum* by its narrower tubular cystidia (40–50 µm × 4–6 µm) [38]; *H. nemorale* is separated from *H. microporoides* by having the colliculose hymenial surface, wider moniliform cystidia (35–70 µm × 7–8 µm) and basidiospores (9.5–14 µm × 4–5 µm) [27]; *H. sinense* differs from *H. membranaceum* by having the encrusted cystidia and smaller basidiospores (8–11.5 µm × 3–5 µm) [25], and another species *H. transiens* differs in its odontioid hymenial surface and narrower basidiospores (9–13 µm × 3–4.5 µm) [39].

Morphologically, *Hyphoderma ayresii* (Berk. ex Cooke) Boidin and Gilles, *H. cremeum* Sheng H. Wu and *H. rimulosum* Sheng H. Wu are similar to *H. crystallinum* by having encrusted cystidia. However, *H. ayresii* differs in its larger encrusted cystidia (70–130 µm × 13–20 µm) and wider basidiospores (9.5–12.5 µm × 6–8 µm) [38]; *H. cremeum* differs from *H. crystallinum* by having both larger encrusted cystidia (40–90 µm × 10–15 µm) and basidia (35–50 µm × 6.5–8 µm) [40]; *H. rimulosum* is separated from *H. crystallinum* by smaller basidiospores (6–7 µm × 3.9–4.1 µm) [41]. *Hyphoderma incrustatum* K.H. Larss., *H. medioburiense*, *H. multicystidium* (Hjortstam and Ryvarden) Hjortstam and Tellería and *H. roseocremeum* (Bres.) Donk are similar to *H. crystallinum* by having tubular cystidia. However, *H. incrustatum* differs from *H. crystallinum* by the porulose hymenial surface and the larger tubular cystidia (50–80 µm × 6–10 µm) [42]; *H. medioburiense* is separated from *H. crystallinum* by the porulose hymenial surface and the larger tubular cystidia (60–100 µm × 7–10 µm) [8]; *H. multicystidium* differs in its larger tubular cystidia (60–80 µm × 5–7 µm), larger basidia (35–50 µm × 5–7 µm) and smaller basidiospores (8–10 µm × 4.5–5 µm) [43]; *H. roseocremeum* differs from *H. crystallinum* by having larger tubular cystidia (80–100 × 6–9 µm) and smaller basidiospores (8–12 µm × 3–4 µm) [8].

*Hyphoderma litschaueri*, *H. moniliforme* (P.H.B. Talbot) Manjón, G. Moreno and Hjortstam, *H. paramacaronesicum* Tellería et al., *H. prosopidis* (Burds.) Tellería et al. and *H. sinense* are similar to *H. membranaceum* by having moniliform or apically moniliform cystidia. However, *H. litschaueri* differs from *H. membranaceum* by having larger moniliform cystidia (60–100 µm × 6–8 µm) and narrower basidiospores (9–12 µm × 3–4 µm) [44]; *H. moniliforme* differs from *H. membranaceum* by having smaller basidiospores (8–9 µm × 3.5–4 µm) [27]; *H. paramacaronesicum* differs in its having both larger moniliform cystidia (70–124 µm × 8–13 µm) and basidia (40−48 µm × 6−9 µm), and wider basidiospores (12–15 µm × 5.5–7 µm) [14]; *H. prosopidis* differs from *H. membranaceum* by the arachnoid to farinaceous hymenial surface and larger basidia (40−45 µm × 8−11 µm) [17]; and *H. sinense* differs in its having encrusted cystidia (18.5–38 µm × 6–11 µm) and smaller basidiospores (8–11.5 µm × 3–5 µm) [25].

*Hyphoderma clavatum* Sheng H. Wu, *H. etruriae* Bernicchia, *H. incrustatum*, *H. orphanellum* (Bourdot & Galzin) Donk, and *H. subclavatum* Sheng H. Wu are similar to *H. microporoides* by having capitate cystidia. However, *H. clavatum* differs from *H. microporoides* by the tuberculate hymenial surface and larger basidiospores (10–13 µm × 4.2–5.2 µm) [41]; *H. etruriae* differs from *H. microporoides* by the grandinioid hymenial surface and wider basidiospores (9–11 µm × 5.5–6.5 µm) [45]; *H. incrustatum* differs in having larger basidiospores (11–14 µm × 4–5 µm) [42]; *H. orphanellum* differs from *H. microporoides* by having larger capitate cystidia (50–80 µm × 8–10 µm) and wider basidiospores (8–10 µm × 5–6 µm) [8]; *H. subclavatum* is separated from *H. microporoides* by having both larger basidia (40–55 µm × 7–8 µm) and basidiospores (10–12 µm × 4.2–5.3 µm) [41].

*Hyphoderma* species are an extensively studied group [10,46], mainly distributed in Europe (e.g., Austria, Russia, France, Germany, Poland, UK, The Netherlands, Portugal, Sweden, Italy, Denmark, Norway, Finland, Spain) (Figure 8) and mainly found on hardwood, although a few species grow on coniferous wood. Many species of *Hyphoderma* were found in Europe, but most of them have not been reported in northern China (Figure 8), in which we presumed that *Hyphoderma* are undersampled by mycologists. Several studies on new wood-decaying fungi of *Hyphoderma* from China have been reported [15,40,41,46], in which 26 *Hyphoderma* species were reported, *H. acystidiatum* Sheng H. Wu, *H. clavatum*, *H. cremeoalbum* (Höhn. and Litsch.) Jülich, *H. cremeum*, *H. definitum* (H.S. Jacks.) Donk, *H. densum* Sheng H. Wu, *H. fissuratum*, *H. floccosum* C.L. Zhao and Q.X. Guan, *H. litschaueri*, *H. crystallinum*, *H. medioburiense*, *H. microcystidium* Sheng H. Wu, *H. microporoides*, *H. moniliforme*, *H. mopanshanense*, *H. nemorale*, *H. obtusiforme* J. Erikss. and Å. Strid, *H. pinicola*, *H. rimulosum*, *H. setigerum*, *H. sibiricum* (Parmasto) J. Erikss. and Å. Strid, *H. sinense*, *H. subclavatum*, *H. subsetigerum* Sheng H. Wu, *H. transiens,* and *H. membranaceum* [8,18,25,27,29,40,41,46]. Further studies should focus on the relationships between the host and *Hyphoderma* species, as well as trying to better understand the evolutionary directions between plant and *Hyphoderma* species. The researches on the phylogeny of *Hyphoderma,* as well as many fungal studies on the molecular systematics [47,48,49], will be useful to push the further research on fundamental research and applied research of fungi. More species of *Hyphoderma* should be found in subtropical and tropical Asia as it was shown that wood-inhabiting fungi are rich in tropical China [50,51].


**Key to 26 accepted species of *Hyphoderma* in China**


Cystidia absent 2Cystidia present 5Hymenial surface grandinioid *H. acystidiatum*Hymenial surface smooth 3Basidiospores > 10.5 µm in length *H. densum*Basidiospores < 10.5 µm in length 4Hymenophore cracked; basidiospores > 8.5 µm in length *H. fissuratum*Hymenophore uncracked; basidiospores < 8.5 µm in length *H. sibiricum*Hymenophore smooth 6Hymenophore tuberculate, porulose, grandinioid, or odontoid 14Two types of cystidia present 7One type of cystidia present 8Moniliform cystidia absent *H. microcystidium*Moniliform cystidia present *H. sinense*Hymenophore uncracked 9Hymenophore cracked 10Basidiospores > 11 µm in length *H. definitum*Basidiospores < 11 µm in length *H. microporoides*Cystidia moniliform 11Cystidia cylindrical 12Basidiospores > 9 µm in length *H. litschaueri*Basidiospores < 9 µm in length *H. moniliforme*Basidiospores ellipsoid < 10 μm in length *H. rimulosum*Basidiospores cylindrical > 10 μm in length 13Basidiospores > 12 µm in length *H. cremeum*Basidiospores < 12 µm in length *H. subclavatum*Hymenophore odontoid or grandinioid 15Hymenophore tuberculate, porulose 16Hymenophore odontoid, basidiospores > 9 µm in length *H. transiens*Hymenophore grandinioid, basidiospores < 9 µm in length *H. subsetigerum*Cystidia of two types 17Cystidia of one type 19Septate cystidia absent *H. crystallinum*Septate cystidia present 18Basidia 2-sterigmata, basidiospores > 13 µm in length *H. pinicola*Basidia 4-sterigmata, basidiospores < 13 µm in length *H. floccosum*Septate cystidia present 20Septate cystidia absent 21Hymenophore porulose to pilose, basidia < 5 µm in width *H. mopanshanense*Hymenophore tuberculate, basidia > 5 µm in width *H. setigerum*Hymenophore porulose *H. obtusiforme*Hymenophore tuberculate, colliculose 22Cystidia > 30 µm in length 23Cystidia < 30 µm in length *H. cremeoalbum*Basidia > 30 µm in length 24Basidia < 30 µm in length 25Hymenophore cracking, cystidia < 10 µm in width *H. medioburiense*Hymenophore not cracking, cystidia > 10 µm in width *H. clavatum*Hymenophore colliculose *H. nemorale*Hymenophore tuberculate *H. membranaceum*

## Figures and Tables

**Figure 1 jof-07-00308-f001:**
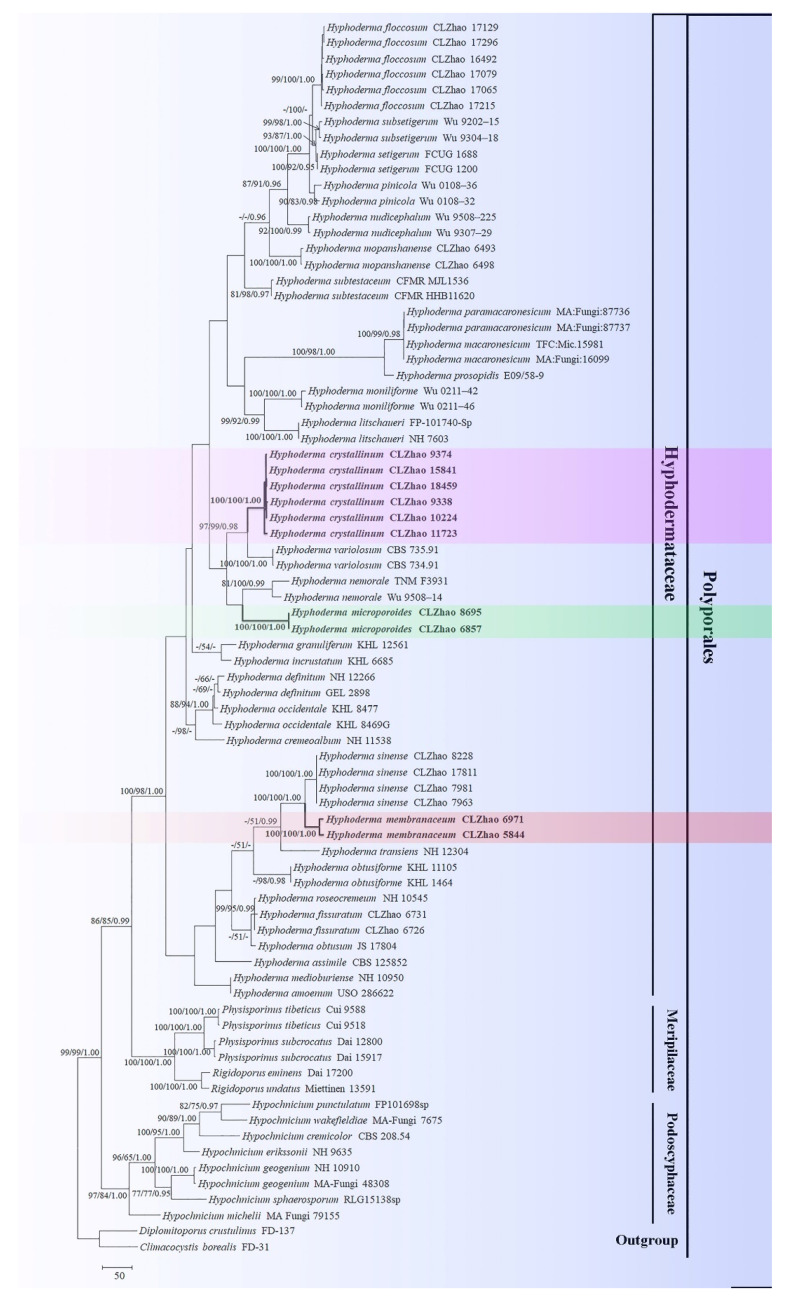
Maximum Parsimony strict consensus tree illustrating the phylogeny of three new species and related species in *Hyphoderma* within Polyporales based on internal transcribed spacer + nuclear ribosomal RNA gene (ITS+nLSU) sequences. Branches are labeled with maximum likelihood bootstrap values > 70%, parsimony bootstrap values > 50% and Bayesian posterior probabilities > 0.95. The new species are in bold. Clade names follow previous study [19].

**Figure 2 jof-07-00308-f002:**
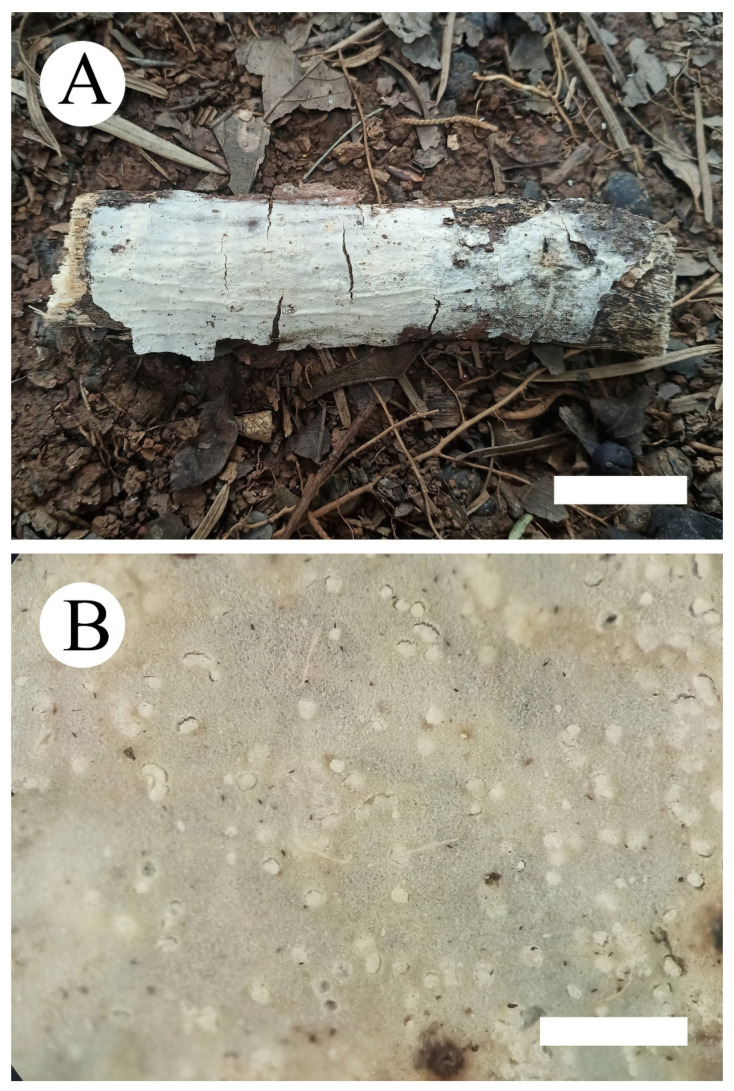
*Hyphoderma crystallinum* (holotype) (**A**): basidiomata on the substrate (**B**); scattered nubby crystals. Bars: **A** = 2 cm and **B** = 1 mm.

**Figure 3 jof-07-00308-f003:**
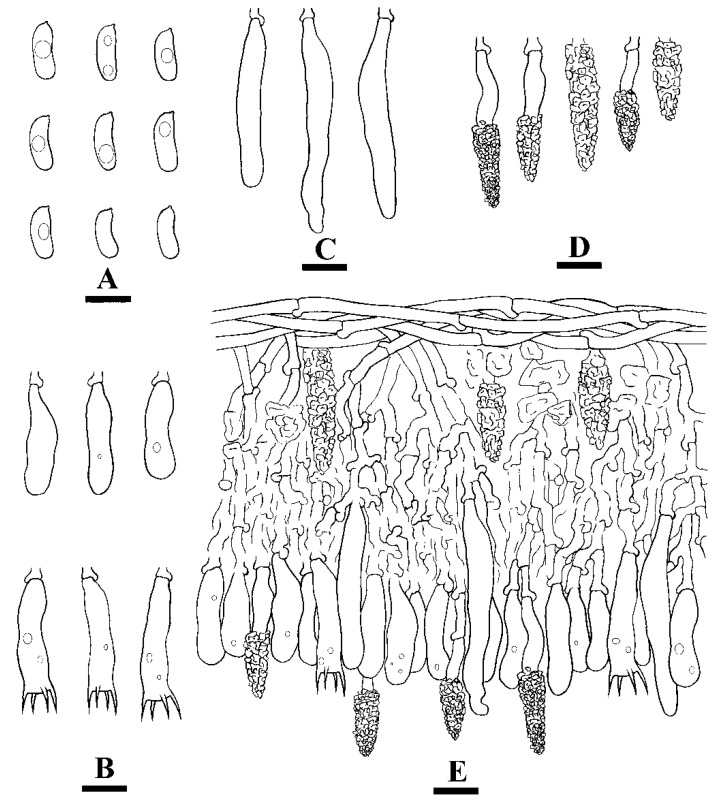
Microscopic structures of *Hyphoderma crystallinum* (holotype) (**A**): basidiospores (**B**), basidia and basidioles (**C**), tubular cystidia (**D**), and encrusted cystidia (**E**). A section of hymenium. Bars: **A**–**E** = 10 µm.

**Figure 4 jof-07-00308-f004:**
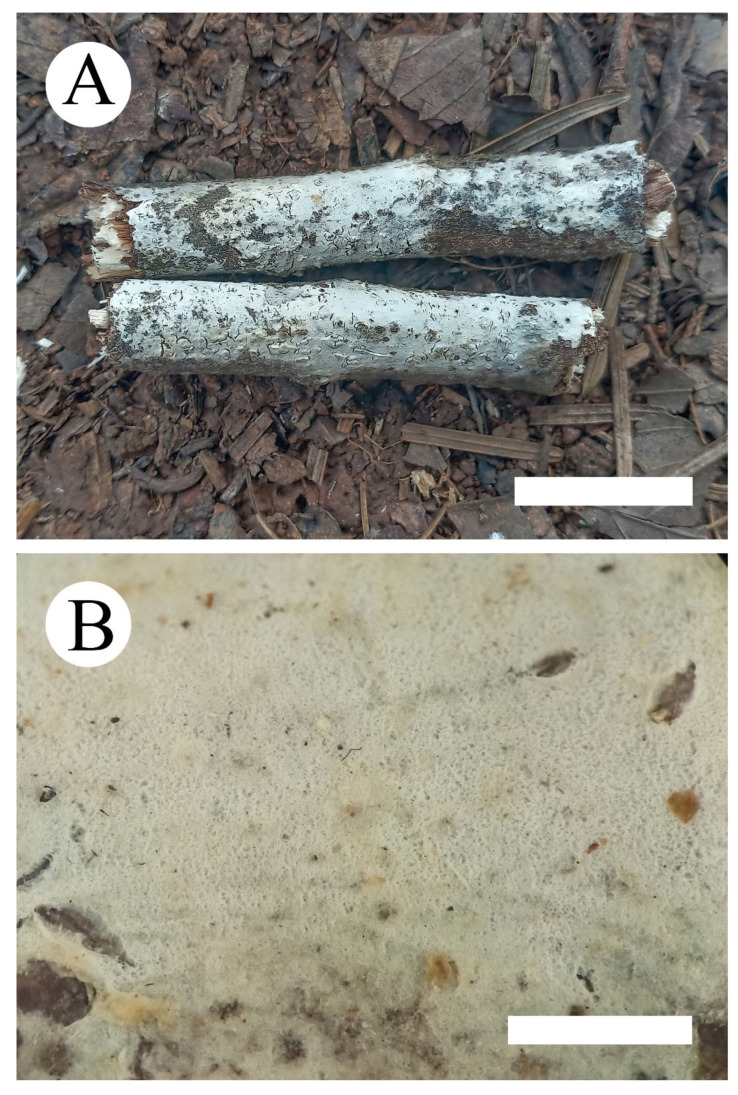
Basidiomata (**A**) of *Hyphoderma membranaceum* (**B**) (holotype). Bars: **A** = 2 cm and **B** = 1 mm.

**Figure 5 jof-07-00308-f005:**
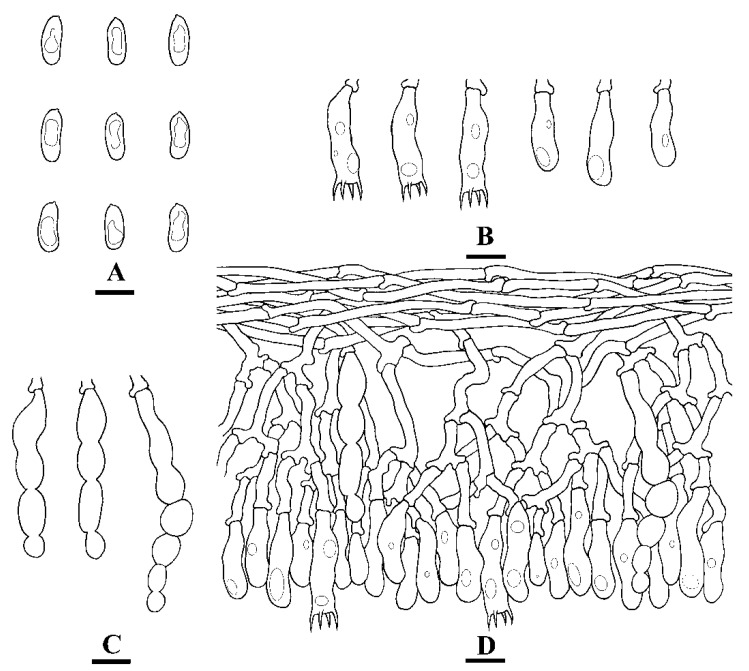
Microscopic structures of *Hyphoderma membranaceum* (holotype) (**A**): basidiospores (**B**), basidia and basidioles (**C**), and cystidia (**D**). A section of hymenium. Bars: **A**–**D** = 10 µm.

**Figure 6 jof-07-00308-f006:**
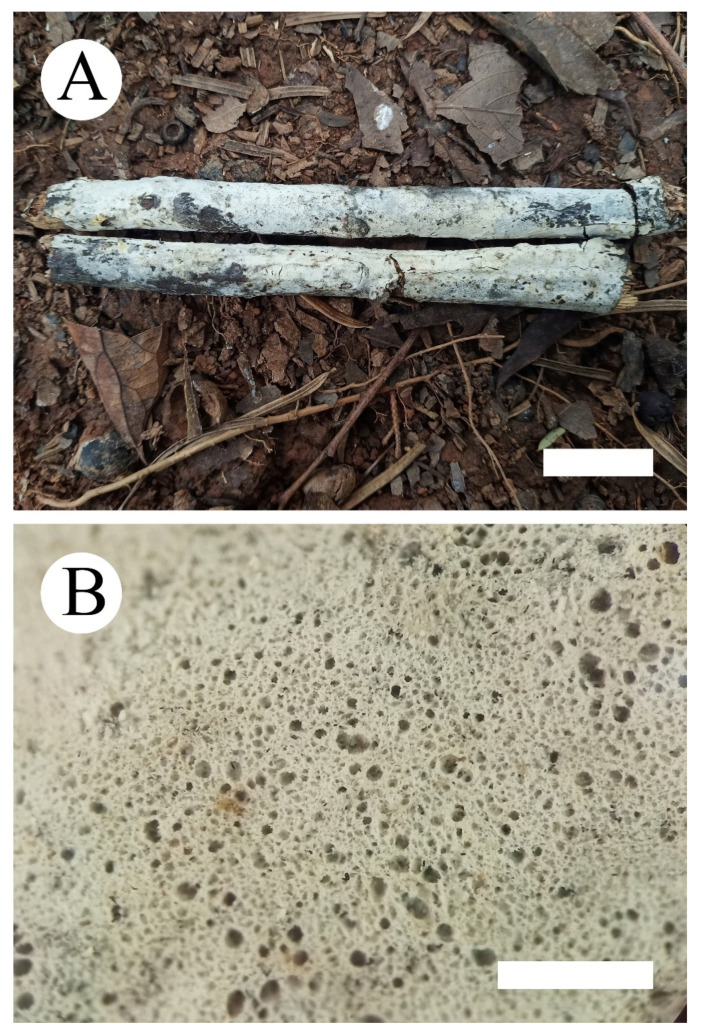
*Hyphoderma microporoides* (holotype) (**A**): basidiomata on the substrate (**B**) and scattered pinholes. Bars: **A** = 2 cm, **B** = 1 mm.

**Figure 7 jof-07-00308-f007:**
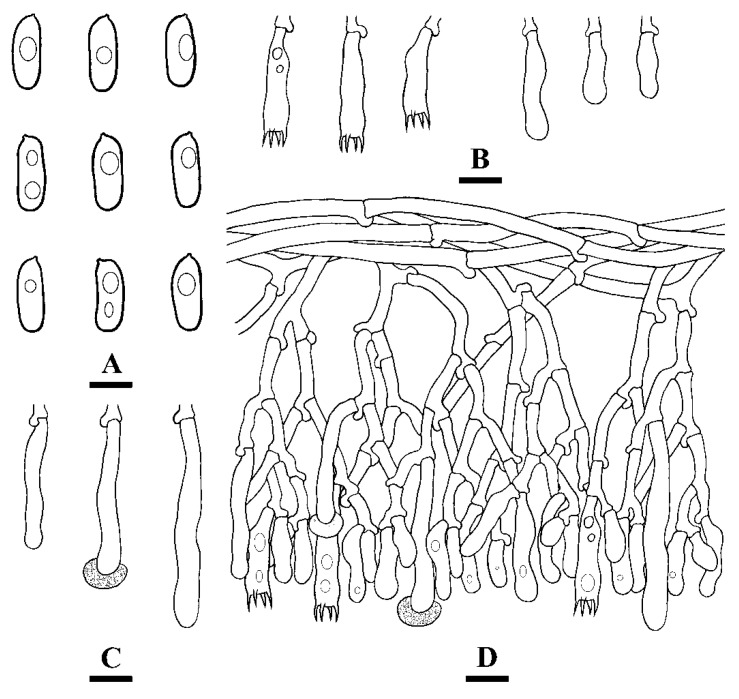
Microscopic structures of *Hyphoderma microporoides* (holotype) (**A**): basidiospores (**B**), basidia and basidioles (**C**), and cystidia (**D**). A section of hymenium. Bars: **A** = 5 µm, **B**–**D** = 10 µm.

**Figure 8 jof-07-00308-f008:**
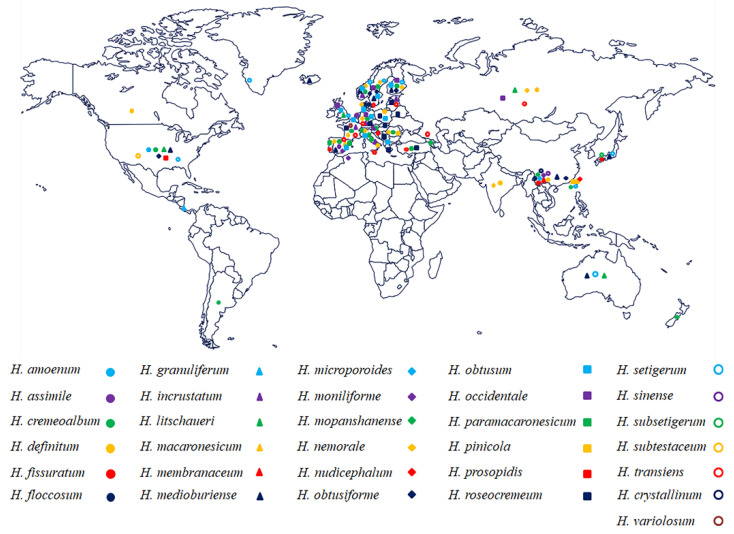
Geographic distribution of *Hyphoderma* species treated in the phylogenetic tree.

**Table 1 jof-07-00308-t001:** List of species, specimens, and GenBank accession numbers of sequences used in this study.

Species Name	Specimen No.	GenBank Accession No.		References
		ITS	LSU	
*Climacocystis borealis*	FD-31	KP135308	KP135210	[19]
*Diplomitoporus crustulinus*	FD-137	KP135299	KP135211	[19]
*Hyphoderma amoenum*	USO 286622	HE577030		[17]
*H. assimile*	CBS 125852	MH863808	MH875272	[24]
*H. cremeoalbum*	NH 11538	DQ677492	DQ677492	[16]
***H. crystallinum***	**CLZhao 9338**	**MW917161**	**MW913414**	**Present study**
***H. crystallinum***	**CLZhao 9374**	**MW917162**	**MW913415**	**Present study**
***H. crystallinum***	**CLZhao 10224**	**MW917163**	**MW913416**	**Present study**
***H. crystallinum***	**CLZhao 11723**	**MW917164**	**MW913417**	**Present study**
***H. crystallinum***	**CLZhao 15841**	**MW917165**	**MW913418**	**Present study**
***H. crystallinum***	**CLZhao 18459**	**MW917166**	**MW913419**	**Present study**
*H. definitum*	GEL 2898		AJ406509	[18]
*H. definitum*	NH 12266	DQ677493	DQ677493	[16]
*H.* *fissuratum*	CLZhao 6731	MT791331	MT791335	[15]
*H.* *fissuratum*	CLZhao 6726	MT791330	MT791334	[15]
*H. floccosum*	CLZhao 17129	MW301683	MW293733	[25]
*H. floccosum*	CLZhao 17296	MW301686	MW293736	[25]
*H. floccosum*	CLZhao 16492	MW301688	MW293734	[25]
*H. floccosum*	CLZhao 17215	MW301687	MW293735	[25]
*H. floccosum*	CLZhao 17079	MW301685		[25]
*H. floccosum*	CLZhao 17065	MW301684		[25]
*H. granuliferum*	KHL 12561	JN710545	JN710545	[18]
*H. incrustatum*	KHL 6685		AY586668	[18]
*H. litschaueri*	NH 7603	DQ677496	DQ677496	[16]
*H. litschaueri*	FP-101740- sp	KP135295	KP135219	[26]
*H. macaronesicum*	MA:Fungi:16099	HE577027		[18]
*H. macaronesicum*	TFC:Mic.15981	HE577028		[18]
*H. medioburiense*	NH 10950	DQ677497	DQ677497	[16]
***H. membranaceum***	**CLZhao 5844**	**MW917167**	**MW913420**	**Present study**
***H. membranaceum***	**CLZhao 6971**	**MW917168**	**MW913421**	**Present study**
***H. microporoides***	**CLZhao 6857**	**MW917169**	**MW913422**	**Present study**
***H. microporoides***	**CLZhao 8695**	**MW917170**	**MW913423**	**Present study**
*H. moniliforme*	Wu 0211-42	KC928282	KC928283	[27]
*H. moniliforme*	Wu 0211-46	KC928284	KC928285	[27]
*H. mopanshanense*	CLZhao 6498	MT791329	MT791333	[15]
*H. mopanshanense*	CLZhao 6493	MT791328	MT791332	[15]
*H. nemorale*	TNM F3931	KJ885183	KJ885184	[27]
*H. nemorale*	Wu 9508-14	KC928280	KC928281	[27]
*H. nudicephalum*	Wu 9307-29	AJ534269		[28]
*H. nudicephalum*	Wu 9508-225	AJ534268		[28]
*H. obtusiforme*	KHL 1464	JN572909		[29]
*H. obtusiforme*	KHL 11105	JN572910		[29]
*H. obtusum*	JS 17804		AY586670	[29]
*H. occidentale*	KHL 8469		AY586674	[29]
*H. occidentale*	KHL 8477	DQ677499	DQ677499	[16]
*H. paramacaronesicum*	MA:Fungi:87736	KC984399	KF150074	[14]
*H. paramacaronesicum*	MA:Fungi:87737	KC984405	KF150073	[14]
*H. pinicola*	Wu 0108-32	KJ885181	KJ885182	[29]
*H. pinicola*	Wu 0108-36	KC928278	KC928279	[29]
*H. prosopidis*	E09/58-9	HE577029		[29]
*H. roseocremeum*	NH 10545		AY586672	[29]
*H. setigerum*	FCUG 1200	AJ534273		[28]
*H. setigerum*	FCUG 1688	AJ534272		[28]
*H. sinense*	CLZhao 7963	MW301679	MW293730	[25]
*H. sinense*	CLZhao 17811	MW301682	MW293732	[25]
*H. sinense*	CLZhao 8228	MW301681		[25]
*H. sinense*	CLZhao 7981	MW301680	MW293731	[25]
*H. subsetigerum*	Wu 9304-18	AJ534277		[28]
*H. subsetigerum*	Wu 9202-15	AJ534278		[28]
*H. subtestaceum*	HHB11620	GQ409521		[29]
*H. subtestaceum*	CFMR MJL1536	GQ409522		[29]
*H. transiens*	NH 12304	DQ677504	DQ677504	[16]
*H. variolosum*	CBS 734.91	MH862320	MH873992	[24]
*H. variolosum*	CBS 735.91	MH862321	MH873993	[24]
*Hypochnicium cremicolor*	CBS 208.54	MH857294	MH868826	[24]
*H. erikssonii*	NH 9635	DQ677508	DQ677508	[16]
*H. geogenium*	NH 10910	DQ677509	DQ677509	[16]
*H. geogenium*	MA-Fungi 48308	FN552534	JN939576	[30]
*H. michelii*	MA-Fungi 79155	NR119742	NG060635	[30]
*H. punctulatum*	FP101698sp	KY948827	KY948860	[19]
*H. sphaerosporum*	RLG15138sp	KY948803	KY948861	[19]
*H. wakefieldiae*	MA-Fungi 7675	FN552531	JN939577	[30]
*Physisporinus subcrocatus*	Dai 15917	KY131870	KY131926	[31]
*P. subcrocatus*	Dai 12800	KY131869	KY131925	[31]
*P. tibeticus*	Cui 9588	KY131873	KY131929	[31]
*P. tibeticus*	Cui 9518	KY131872	KY131928	[31]
*Rigidoporus eminens*	Dai 17200	MT279690	MT279911	[31]
*R. undatus*	Miettinen-13591	KY948731	KY948870	[19]

New sequences are shown in bold.

## Data Availability

Publicly available datasets were analyzed in this study. This data can be found here: [https://www.ncbi.nlm.nih.gov/; https://www.mycobank.org/page/Simple%20names%20search; http://purl.org/phylo/treebase, submission ID 27983; accessed on 16 April 2021].

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
