# Peer review of "Taxonomy and Phylogeny of the Wood-Inhabiting Fungal Genus Hyphoderma with Descriptions of Three New Species from East Asia"

_jof, 2021, doi:10.3390/jof7040308_

Round 1
Reviewer 1 Report
This paper deals with three new wood-decaying fungi in the genus Hyphoderma from Yunnan Province of SW China. A detailed description, photo and drawing for each species are provided. In addition, phylogenetic analysis on Hyphoderma was carried out. These data are novel and worth to be published.
However, the manuscript needs to be revised:
1) The English should be polished by native speaker.
2) The "macrocrystallina" and "vacuspora" are not good, even not correct.
3) Line 322"China are poorly reported [46], in which twenty-six Hyphoderma species were reported". this is contradiction:poorly studied, why 26 species were found in China?
4) In Figure 6, many species of Hyphoderma were found in Europe, why almost no records in northern China? the authors should discuss this problem.
5) I believe more species of Hyphoderma occur in subtropical and tropical Asia, because polypores are a well study group of wood-inhabiting fungi, and they are much rich in tropical China (Wu et al. 2020), it is very possible the same phenomenon for Hyphoderma.
Wu F, Yuan HS, Zhou LW, Yuan Y, Cui BK, Dai YC, 2020. Polypore diversity in South China. Mycosystema 39: 653–681.
Author Response
Dear Editor,
We are very grateful to you for your patient comments on our manuscript. We have carefully revised the manuscript directly in the text in red color, according to these comments.
The responses to the comments were listed below and highlighted in green color.
Warm regards,
Qian-Xin Guan, Chang-Lin Zhao
---------------------------------------------
Reviewers' comments:
This paper deals with three new wood-decaying fungi in the genus Hyphoderma from Yunnan Province of SW China. A detailed description, photo and drawing for each species are provided. In addition, phylogenetic analysis on Hyphoderma was carried out. These data are novel and worth to be published.
The notes for revision:
1) The English should be polished by native speaker.
Response: We have invited Julie McIntosh Shapiro to polish this manuscript (julie_shapiro@harvard.edu; Office: 617-495-1029; Harvard University).
2) The "macrocrystallina" and "vacuspora" are not good, even not correct.
Response: We have revised them as “Hyphoderma crystallinum”and “Hyphoderma membranaceum”according to the reviewer's comments.
3) Line 322"China are poorly reported [46], in which twenty-six Hyphoderma species were reported". This is contradiction: poorly studied, why 26 species were found in China?
Response (Line 318-319): We have revised this sentence as “The researches on the new taxa related with wood-decaying fungi of Hyphoderma from China have been reported [15,40,41,46]”.
4) In Figure 8, many species of Hyphoderma were found in Europe, why almost no records in northern China? The authors should discuss this problem.
Response (Line 315-318): We have revised them as “Many species of Hyphoderma were found in Europe, but most of them did not record in northern China in Figure 8, in which we presumed that the specimens of Hyphoderma were not collected by the mycologists”.
5) I believe more species of Hyphoderma occur in subtropical and tropical Asia, because polypores are a well study group of wood-inhabiting fungi, and they are much rich in tropical China (Wu et al. 2020), it is very possible the same phenomenon for Hyphoderma.
Response (Line 331-334): We have revised it as “I believe more species of Hyphoderma occur in subtropical and tropical Asia, because polypores are a well study group of wood-inhabiting fungi, and they are much rich in tropical China [50,51], and it is very possible the same phenomenon for Hyphoderma”.
Reviewer 2 Report
Title –‘’Taxonomy and phylogeny of the forest ecological fungal genus Hyphoderma with descriptions of three new wood-environmental species from East Asia’’The title is rather long. What about ‘’Taxonomy and phylogeny of the wood-inhabiting fungal genus Hyphoderma with descriptions of three new species from East Asia’’Line 18 –I would delete spore measurements from abstract. ‘’(11–13.5 × 4.5–5.5 μm)’’Line 23 –ditto for ‘’(100% BS, 100% BP and 1.00 BPP)’’Line 28-44 –The first paragraph reads as though it is directly copied from the given references. It needs to be shortened and written in your own words.Lines 45-52 –this is a very long sentence. Break into 2 or 3 sentences for better understanding and ease of reading.Line 63 –‘’grouped overlapped and it proposed a new species’’ I suggest ‘’grouped and they proposed a new species’’Line 62/63 –do you need to give all of the authorities for H. macaronesicum, or can it be reduced to Tellería et al.?Also Line 306 -H. paramacaronesicum Telleria, M. Dueñas, J. Fernández-López & M.P. MartínLine 70-72 –sentence needs re-writing to make clear and improve English. Suggest ‘’Chinese Hyphoderma species were compared with closely related taxa and two new species were proposed, H. fissuratum C.L. 72 Zhao & X. Ma and H. mopanshanense C.L. Zhao [15].Line 73-74 –Suggest, ‘’We collected material of three undescribed species of corticioid fungi from Yunnan Province, P.R. China.’’Line 75-77 –Suggest, ‘’We present morphological and molecular phylogenetic evidence that supports the recognition of three new species within Hyphoderma, based on the internal transcribed spacer (ITS) regions nLSU sequences.’’Line 85 –‘’are used’’, not ‘’were used’’Line 93-‘’instructions following a previous study’’, not ‘’instructions followed previous study’’Line 113 –‘’following’’, not ‘’followed’’Line 169 –‘’becominghard’’, not‘’turn to hard’’Line 171 –‘’scattered nubby crystals’’, not ‘’scattering nubby crystal’’. Also, is this what is shown in Fig. 2b? If so, then indicate in figure caption.Line 210 –‘’Hymenial surface smooth with scattered pinholes visible using hand lens,’’, not ‘’Hymenial surface smooth and distributing the tiny pinhole under the lens,’’. Also, is this whatis shown in Fig. 4b? If so, then indicate in figure caption.Line 211 –add comma after ‘’Margin sterile, ‘’
Line 257-258 ‘’Figure 8. Geographic distribution of Hyphoderma species treated in the phylogenetic tree.’’, not ‘’Figure 8. The map of geographic distribution of Hyphoderma in the phylogenetic tree worldwide.’’Line 265 –‘’species’’, not ‘’specieof’’Line 272 –delete ‘’both’’Line 279 –‘’by having encrusted cystidia’’, not ‘’on the basis of the character by having the encrusted cystidia’’Line 286 –‘’by having tubular cystidia’’, not ‘’on the basis of the character by having tubular cystidia’’Line 292 –delete ‘’the’’Line 296 –‘’by having capitate cystidia’’, not ‘’on the basis of the character by having the capitate cystidia’’Line 300–delete ‘’its’’ Line 301, 309, 311, 315–delete ‘’the’’Line 308 –‘’by having moniliform’’, not ‘’on the basis of the character by having the moniliform’’Line 317-320 –‘’Hyphoderma species are an extensively studied group [10,46], mainly distributed in Europe (e.g., Austria, Russia, France, Germany, Poland, UK, Netherlands, Portugal, Sweden, Italy, Denmark, Norway, Finland, Spain) (Fig. 8) and mainly found on hardwood, although a few species grow on coniferous wood.’’, not ‘’In the habitatand distribution, Hyphoderma species are an extensively studied group [10,46], which mainly distribute in Europe, such as Austria, Russia, France, Germany, Poland, UK, Netherlands, Portugal, Sweden, Italy, Denmark, Norway, Finland, Spain 320 (Fig. 8) and the substrata most are hardwood, but few species also on coniferous wood.’’Line 321 –I would delete ‘’The researches on the new taxa related with wood-decaying fungi of Hyphoderma from China are poorly reported [46], in which’’ Then say ‘’Twenty-six species of Hyphoderma have been reported from China.Line 329 –Perhaps say ‘’should’’ rather than ‘’maybe’’Line 335 –‘’Key to 26 accepted species of Hyphoderma in China’’, not ‘’An identification key to 26 accepted species in China’’General –I find that so many authorities are given for the various Hyphoderma species that it makes reading of the manuscript difficult (especially in the discussion). Are they all needed, or is it journal policy? Could they be given in the table instead?
Author Response
Dear Editor,
We are very grateful to you for your patient comments on our manuscript. We have carefully revised the manuscript directly in the text in red color, according to these comments.
The responses to the comments were listed below and highlighted in green color.
Warm regards,
Qian-Xin Guan, Chang-Lin Zhao
---------------------------------------------
Reviewers' comments:
Title: Revised “Taxonomy and phylogeny of the forest ecological fungal genus Hyphoderma with descriptions of three new wood-environmental species from East Asia” as “Taxonomy and phylogeny of the wood-inhabiting fungal genus Hyphoderma with descriptions of three new species from East Asia”;
Response: We have revised it according to the reviewer's comment.
Abstract:
Line 18 –) I would delete spore measurements from abstract. “(11–13.5 × 4.5–5.5 µm)”;
Response (Line 16): We have deleted the spore measurements from abstract.
Line 23 –) ditto for “(100% BS, 100% BP and 1.00 BPP)”;
Response (Line 23): We have deleted it according to the reviewer's comments.
Introduction
Line 28-44 –) The first paragraph reads as though it is directly copied from the given references. It needs to be shortened and written in your own words.
Response (Line 28-45): We have revised them according to the reviewer's comments.
Lines 45-52 –) this is a very long sentence. Break into 2 or 3 sentences for better understanding and ease of reading.
Response (Line 46-53): We have revised it into 3 sentences as “The genus Hyphoderma Wallr. is very important group of wood-environmental fungi because of its key role in the carbon cycle and being the most efficient wood decayers in the forest ecological system [7]. This genus is typified by H. setigerum (Fr.) Donk [8]. Hyphoderma is characterized by the resupinate to effuse-reflexed basidiomata with ceraceous consistency, and smooth to tuberculate or hydnoid hymenophore and a monomitic hyphal structure (rarely dimitic) with clamp connections on generative hyphae, presence of cystidia or not, basidia suburniform to subcylindrical and cylindrical, ellipsoid to subglobose, smooth, thin-walled basidiospores [9,10]”.
Line 63 –) “grouped overlapped and it proposed a new species” I suggest “grouped and they proposed a new species”.
Response (Line 63): We have revised it as “……in which some species from Hyphoderma and Peniophorella grouped and they proposed a new species, H. macaronesicum……”according to the reviewer's comment.
Line 62/63 –) do you need to give all of the authorities for H. macaronesicum, or can it be reduced to Tellería et al.? Also Line 306 - H. paramacaronesicum Telleria, M. Dueñas, J. Fernández-López & M.P. Martín.
Response (Line 63/292): We have revised them as “H. macaronesicum Tellería et al. and H. paramacaronesicum Tellería et al.” according to the reviewer's comments.
Line 70-72 –) sentence needs re-writing to make clear and improve English. Suggest “Chinese Hyphoderma species were compared with closely related taxa and two new species were proposed, H. fissuratum C.L. Zhao & X. Ma and H. mopanshanense C.L. Zhao [15]”.
Response (Line 70-72): We have revised it according to the reviewer's comment.
Line 73-74 –) Suggest, “We collected material of three undescribed species of corticioid fungi from Yunnan Province, P.R. China.”
Response (Line 73-74): We have revised it according to the reviewer's comment.
Line 75-77 –) Suggest, “We present morphological and molecular phylogenetic evidence that supports the recognition of three new species within Hyphoderma, based on the internal transcribed spacer (ITS) regions nLSU sequences.”
Response (Line 74-76): We have revised it.
Materials and Methods
Line 85 –) “are used”, not “were used”.
Response (Line 84): We have revised “were used” as “are used”.
Line 93 –) “instructions following a previous study”, not “instructions followed previous study”.
Response (Line 92): We have revised “instructions followed previous study” as “instructions following a previous study”.
Line 113 –) “following”, not “followed”.
Response (Line 112): We have revised “followed” as “following” according to the reviewer's comment.
Results
Line 169 –) “becoming hard”, not “turn to hard”.
Response (Line 170): We have revised “turn to hard” as “becoming hard”.
Line 171 –) “scattered nubby crystals”, not “scattering nubby crystal”. Also, is this what is shown in Fig. 2b? If so, then indicate in figure caption.
Response (Line 172/167-168): We have revised it as “scattered nubby crystals”. We have indicated in figure caption as “A. Basidiomata; B. scattered nubby crystals”in Fig. 2b.
Line 210 –) “Hymenial surface smooth with scattered pinholes visible using hand lens,”, not “Hymenial surface smooth and distributing the tiny pinhole under the lens,”. Also, is this what is shown in Fig. 4b? If so, then indicate in figure caption.
Response (Line 238-239/229-230): We have revised it as “Hymenial surface smooth with scattered pinholes visible using hand lens”. We have indicated in figure caption as “A. Fruiting body; B. Scattered pinholes”in Fig. 4b.
Line 211 –) add comma after “Margin sterile,”
Response (Line 239): We have revised it according to the reviewer's comment.
Line 257-258 –) “Figure 8. Geographic distribution of Hyphoderma species treated in the phylogenetic tree.” not “Figure 8. The map of geographic distribution of Hyphoderma in the phylogenetic tree worldwide.”
Response (Line 257): We have revised it according to the reviewer's comment.
Discussion
Line 265 –) “species”, not “specie of”
Response (Line 264): We have revised “specie of” as “species” according to the reviewer's comment.
Line 272 –) delete “both”
Response (Line 271): We have deleted it.
Line 279 –) “by having encrusted cystidia”, not “on the basis of the character by having the encrusted cystidia”
Response (Line 277-278): We have revised it according to the reviewer's comment.
Line 286 –) “by having tubular cystidia”, not “on the basis of the character by having tubular cystidia”
Response (Line 284): We have revised it.
Line 292 –) delete “the”
Response (Line 290): We have deleted it.
Line 296 –) “by having capitate cystidia”, not “on the basis of the character by having the capitate cystidia”
Response (Line 305): We have revised it according to the reviewer's comment.
Line 300 –) delete “its”
Response (Line 308): We have deleted it.
Line 294, 296, 300, 308-309) delete “the”
Response (Line 295, 297, 301, 309-310): We have deleted them.
Line 308 –) “by having moniliform”, not “on the basis of the character by having the moniliform”
Response (Line 294): We have revised it according to the reviewer's comment.
Line 317-320 –) “Hyphoderma species are an extensively studied group [10,46], mainly distributed in Europe (e.g., Austria, Russia, France, Germany, Poland, UK, Netherlands, Portugal, Sweden, Italy, Denmark, Norway, Finland, Spain) (Fig. 8) and mainly found on hardwood, although a few species grow on coniferous wood.”, not “In the habitat and distribution, Hyphoderma species are an extensively studied group [10,46], which mainly distribute in Europe, such as Austria, Russia, France, Germany, Poland, UK, Netherlands, Portugal, Sweden, Italy, Denmark, Norway, Finland, Spain 320 (Fig. 8) and the substrata most are hardwood, but few species also on coniferous wood.”
Response (Line 313-316): We have revised it.
Line 321 –) I would delete “The researches on the new taxa related with wood-decaying fungi of Hyphoderma from China are poorly reported [46], in which” Then say “Twenty-six species of Hyphoderma have been reported from China”.
Response (Line 319-320): We have revised it as “The researches on the new taxa related with wood-decaying fungi of Hyphoderma from China have been reported [15,40,41,46], in which twenty-six Hyphoderma species were reported”.
Line 329 –) Perhaps say “should” rather than “maybe”
Response (Line 328): We have revised “maybe” as “should”.
Line 335 –) “Key to 26 accepted species of Hyphoderma in China”, not “An identification key to 26 accepted species in China”
Response (Line 336): We have revised it according to the reviewer's comment.
General –) I find that so many authorities are given for the various Hyphoderma species that it makes reading of the manuscript difficult (especially in the discussion). Are they all needed, or is it journal policy? Could they be given in the table instead?
Response: There is a good idea. I agree with you. However, it is journal policy to have this part.